

# Customized deep learning based Turkish automatic speech recognition system supported by language model

Yasin Görmez

Management Information System, Sivas Cumhuriyet University, Sivas, Merkez, Turkiye

## ABSTRACT

**Background**. In today's world, numerous applications integral to various facets of daily life include automatic speech recognition methods. Thus, the development of a successful automatic speech recognition system can significantly augment the convenience of people's daily routines. While many automatic speech recognition systems have been established for widely spoken languages like English, there has been insufficient progress in developing such systems for less common languages such as Turkish. Moreover, due to its agglutinative structure, designing a speech recognition system for Turkish presents greater challenges compared to other language groups. Therefore, our study focused on proposing deep learning models for automatic speech recognition in Turkish, complemented by the integration of a language model.

**Methods**. In our study, deep learning models were formulated by incorporating convolutional neural networks, gated recurrent units, long short-term memories, and transformer layers. The Zemberek library was employed to craft the language model to improve system performance. Furthermore, the Bayesian optimization method was applied to fine-tune the hyper-parameters of the deep learning models. To evaluate the model's performance, standard metrics widely used in automatic speech recognition systems, specifically word error rate and character error rate scores, were employed.

**Results**. Upon reviewing the experimental results, it becomes evident that when optimal hyper-parameters are applied to models developed with various layers, the scores are as follows: Without the use of a language model, the Turkish Microphone Speech Corpus dataset yields scores of 22.2 -word error rate and 14.05-character error rate, while the Turkish Speech Corpus dataset results in scores of 11.5 -word error rate and 4.15 character error rate. Upon incorporating the language model, notable improvements were observed. Specifically, for the Turkish Microphone Speech Corpus dataset, the word error rate score decreased to 9.85, and the character error rate score lowered to 5.35. Similarly, the word error rate score improved to 8.4, and the character error rate score decreased to 2.7 for the Turkish Speech Corpus dataset. These results demonstrate that our model outperforms the studies found in the existing literature.

Corresponding author
Yasin Görmez, ysngrmz@hotmail.com

# INTRODUCTION

Today, voice command applications are used in various systems, including smart home systems, smart assistants, in-car systems, and customer services. In a voice command application, the sound captured by a microphone needs to be converted into a format that the application can comprehend. To achieve this interpretation, the obtained sound recordings must be categorized into classes or converted into character strings. These systems, known as automatic speech recognition (ASR), are extensively researched today thanks to the advancement of artificial intelligence. In particular, techniques that convert a given sound into a string, known as sequence-to-sequence (seq2seq) methods, are gaining popularity (*Chiu et al., 2018*). The purpose of seq2seq models is to transform a given input sequence of information into another output sequence. The most significant distinction between seq2seq models and classical classification problems is that in seq2seq models, the lengths of both the input and output sequences can vary from one sample to another. Hence, seq2seq models can be characterized as more intricate compared to classical classification problems. Considering this information, the development of a high-performance ASR system using the seq2seq method will make significant contributions to various fields.

As seq2seq models are employed for ASR, they involve a more complex process compared to the classical classification problems. Moreover, there are additional challenges that must be addressed when developing ASR systems. Foremost among these challenges involves various factors such as characters, words, and accents across different language groups. Thus, there might be a need to retrain models for distinct language groups and even create new models tailored to the specific characteristics of each language group (*Arora & Singh, 2012*). Another crucial challenge in ASR systems is the presence of interpersonal accent variations. Within the languages, different regional accents can exist, and individuals within the same region may use different accents (*Arora & Singh, 2012*). It is highly improbable to have voice recordings of every individual in the datasets used for training ASR systems. In reality, the datasets used for model training are significantly smaller than the actual user population. Given these circumstances, an ASR model should be designed to generalize effectively and recognize a wide range of voices even with limited data. In this context, employing a dataset rich in accent diversity will enhance the generalization capabilities of the designed model (*Cayir & Navruz, 2021*). Therefore, our study aimed to utilize extensive datasets encompassing participants from various accent groups.

Based on the literature, it is evident that the existing ASR has some challenges. Given that these challenges vary across different languages, designing an ASR system, especially for agglutinative languages like Turkish, is notably more complex than other languages. The structure of the Turkish language allows for the generation of billions of different morphological forms (*Palaz et al., 2005*; *Oyucu & Polat, 2023*). In Turkish, which is in the agglutinative language group, the variety of words is significantly greater compared to languages such as English. This is due to the placement of suffixes at the end of words. For instance, the word "korkusuzlaştırılmış" formed using the word "korku" which is defined as "fear" in English, is expressed as "One who has been made fearless" in English. In Turkish, numerous structures expressed as phrases or sentences in English can be

condensed into single words, as exemplified in this particular case. Hence, developing an ASR system for Turkish requires high-dimensional and a large volume of data, which can be a costly endeavor to acquire. Thus, researchers have turned to approaches that include language models for Turkish ASR systems with lower costs.

Considering all these situations, within the study's scope, a language model was developed to enhance the performance of the proposed models. Upon reviewing the literature, it becomes evident that there are few studies dedicated to the Turkish language, with the majority focusing on languages like English, Chinese, Spanish, German, and French (*Arslan & Barışcı, 2020*). In addition to the complexity of the language, the limited availability of studies also results in a lack of accessible resources. For this reason, it is necessary to design models that will achieve high performance in Turkish ASR systems with few resources. In this study, novel deep learning models were developed for Turkish ASR using convolutional neural networks (CNN), long short-term memories (LSTM), Transformers and gated recurrent units (GRU). Connectionist temporal classification (CTC) was employed as the loss function in these models, and their hyper-parameters were optimized using the Bayesian optimization technique. In this context, the primary contribution of the study to the literature is regarded as twofold: the creation of deep learning models tailored to the Turkish language and the introduction of systematic methodologies for hyper-parameter optimization within this domain.

To reduce the word error rate (WER) of deep learning methods designed for seq2seq predictions, a language model created with the use of the Zemberek (*Zemberek-NLP, 2023*) library was incorporated into the methods. The novelty of the model has also been augmented due to the inclusion of the language model, aimed at enhancing the performance of the model. The Zemberek library was chosen because it is frequently preferred for Turkish text pre-processing (*Akın, Demir & Doğan, 2012*; *Kaya, Fidan & Toroslu, 2012*; *Polat & Oyucu, 2020*; *Toraman et al., 2023*). To the best of our knowledge, the final model specifically developed for the Turkish language within the study's scope is not documented in existing literature. Due to this reason, it is expected that the study will provide a novel model, contributing to the existing literature.

In the subsequent sections, the article proceeds with a comprehensive literature review, followed by an exposition of the materials and methods utilized, the presentation of experimental findings, a discussion of their implications, and concludes with a summary of the study's findings and potential avenues for future research.

## LITERATURE REVIEW

Hidden Markov models (HMMs) were frequently used for ASR, in the past due to their ease of implementation, usability, and inherent compatibility with seq2seq models (*Juang & Rabiner, 1991*). In these systems, the speech signal was considered as a piecewise stationary signal, or in other terms, a short-time stationary signal. These models can be quite inefficient, particularly for short-time stationary signal (*Nassif et al., 2019*). For this reason, in recent years, both classical models and deep learning approaches have been frequently employed in ASR, akin to their application in numerous other problem domains.

*Nguyen, Heigold & Zweig (2010)* achieved an improvement of up to 3% in the sentence error rate for the English language with the proposed flat direct model. In their study, *Abdel-Hamid et al. (2014)* demonstrated that there was an improvement of up to 10% compared to deep neural networks when using CNN on TIMIT phone recognition and the voice search large vocabulary speech recognition datasets. *Rao, Sak & Prabhavalkar (2017)* achieved WER of 8.5% and 5.2% for tasks voice-search and voice-dictation, respectively, by using the proposed recurrent neural network transducer based model. *Liu et al. (2018)* compared LSTM with deep neural networks, CNNs, and bidirectional LSTM models, demonstrating that the LSTM model outperformed the others on the Xiaomi speaker test set. *Toshniwal et al. (2018)* designed an automatic speech system supporting nine different Indian languages using the encode-decode approach. *Chiu et al. (2018)* achieved a 3.6% improvement in WER on a 12,500 h speech task by using the model they developed, which incorporated a unidirectional LSTM encoder layer. *Wang, Wang & Lv (2019)* provided a detailed comparison of the advantages and disadvantages of the popular connectionist temporal classification based, recurrent neural network transducer and attention-based approaches for ASR. *Wang, Wang & Lv (2019)* mentioned that ASR developer did not share the dataset for the Mandarin language and obtained a 19.2% WER score with the CNN, LSTM, and CTC based on model that they trained by using the dataset that they shared as open source in the same study.

*Kamper, Matusevych & Goldwater (2020)* designed a system that could operate in six languages without labeled data by training a model using data from seven languages where labeled data were sufficient. *Hsu et al. (2021)* achieved 13% relative WER reduction on Librispeech dataset using Hidden-Unit Bidirectional Encoder Representations from Transformers (BERT) based deep model. *Tombaloğlu & Erdem (2021)* combined a sub-word based language model and a recurrent units-based ASR model for Turkish language and they achieved 10.65% WER score with the model that used LSTM as a recurrent unit. *Korkmaz & Boyacı (2022)* modified a speech recognition system to predict the speaker's region through dialect information. *Oyucu & Polat (2023)* showed that supporting the ASR model with a language model decreased the WER score, especially for languages with limited resources. *Yu et al. (2022)* combined the biLSTM layer with dimension reduction and showed that they saved up to 0.5 days of processing time on the dataset they analyzed. *Ren et al. (2022)* observed a decrease in the WER score for the LibriSpeech, Common Voice-Turkish, and Common Voice-UZBEK datasets in the ratios of 2.96%, 7.07%, and 7.08%, respectively, by using the proposed feature extraction technique. *Oruh, Viriri & Adegun (2022)* achieved a 99.36% accuracy on the English digit dataset with the model that they proposed to address the memory bandwidth problem of the LSTM layer. *Reza et al. (2023)* obtained 4.7% and 3.61%-character error rate (CER) on the LibriSpeech corpus and LJ Speech datasets, respectively, with the use of residual convolution neural network and bi-directional gated recurrent units- based deep model. *Mussakhojayeva et al. (2023)* demonstrated that their multilingual model for Turkic languages improved automatic speech recognition performance.

It is recognized that the ASR studies targeting languages within the agglutinative group may require distinct approaches compared to the languages in other linguistic

groups. *Kurimo et al. (2006)* argued that employing a word-based ASR system might not be suitable for languages in the agglutinative group. They proposed the utilization of a word-independent ASR system as a more effective alternative in such linguistic contexts. *Mamyrbayev et al. (2020)* devised an end-to-end ASR system utilizing LSTM for the Kazakh language, belonging to the agglutinative language group. Their system yielded WER and CER scores of 8.01 and 17.91, respectively. *Xu, Pan & Yan (2016)* formulated an ASR system targeting Korean and Uyghur languages from the agglutinative group, along with Mandarin, a non-agglutinative language. They employed the automatic allophone derivation method in their design. Their findings revealed that the method based on automatic allophone derivation resulted in an approximate 10% enhancement in the CER score specifically for languages within the agglutinative group (*Xu, Pan & Yan, 2016*). *Valizada (2021)* conducted a comparison between syllable-based sub-word and word-based methods for the Azerbaijani language, situated within the agglutinative group. The analyses conducted within the study revealed that syllable-based sub-word methods achieved a 5% better WER score compared to word-based methods (*Valizada, 2021*). *Guo, Yolwas & Slamu (2023)* succeeded in reducing the number of model parameters and overall model size without compromising model performance for the Kazakh language in the agglutinative group, utilizing their low-level multi-head self-attention encoder and decoder model.

As a summary of the literature review, while HHMs were previously frequently used for ASR, seq2seq artificial intelligence models are now more preferred. It has been found that deep learning methods are commonly employed in recent studies on ASR systems. It is observed that these studies commonly incorporate layers such as GRU, LSTM, and Transformer, resulting in the most successful outcomes among the methods utilized. It has been noted that a majority of ASR studies in existing literature primarily concentrate on languages such as English, Chinese, Spanish, German, and French, whereas there is a scarcity of research focusing on languages within the agglutinative language group, such as Turkish. In addition to all these, it has been concluded that there are various difficulties in ASR systems for languages in the agglutinative group, such as Turkish, compared to other languages, and that using a language model positively affects the model performance.

## MATERIALS & METHODS

### Datasets

In this study, the Turkish Microphone Speech Corpus (METU-1.0), which was prepared with the cooperation of the Middle East Technical University and the Center for Spoken Language Research (*Salor et al., 2002*; *Salor et al., 2007*) and the Turkish Speech Corpus (TSC) (*Mussakhojayeva et al., 2023*) were used as speech datasets. METU-1.0 comprises a total of 5.6 h and 4,769 audio recordings from 193 different participants, whereas TSC includes a total of 218.24 h of audio recordings spanning 186,170 utterances. For detailed information about the datasets, please refer to the respective references provided for each dataset.

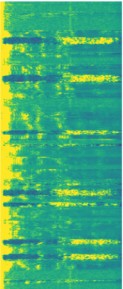
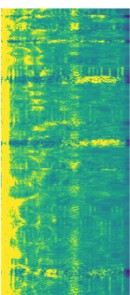

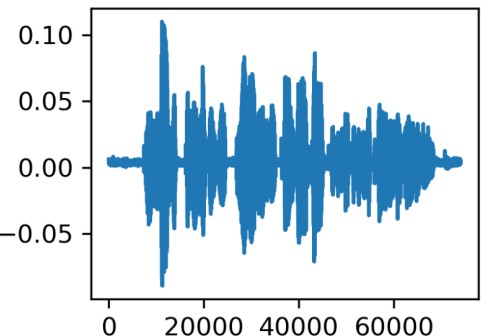
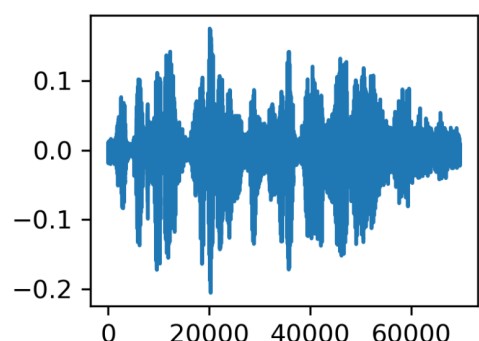

**Figure 1** **The signal waveform, spectrogram, and file label resulting from STFT for samples selected from the datasets.** The figure on the top left is STFT Spectrogram of sound labeled as "gösterdikleri sığınak neredeyse tam radyoaktif saçılma korumasına sahipti". The figure on the bottom left is Signal Wave of sound labeled as "gösterdikleri sığınak neredeyse tam radyoaktif saçılma korumasına sahipti". The figure on the top right is STFT Spectrogram of sound labeled as "çok geçmeden sebebinin kendisini bekleyen yavrularıolduğu anlaşılıyor". The figure on the bottom right is Signal Wave of sound labeled as "çok geçmeden sebebinin kendisini bekleyen yavrularıolduğu anlaşılıyor".

## Feature extraction

Feature extraction is a crucial stage in ASR systems, as in many machine learning applications. In this context, obtaining spectrograms from audio files for use in speech recognition systems enhances system performance. In this study, spectrograms of audio files were obtained using the short-time Fourier transform (STFT). The Fourier analysis provides energy/power separation of signal frequency components but lacks time information, thus not indicating the time period of each frequency component (*Ari, Ayaz & Hanbay, 2019*). On the other hand, the STFT calculates the Fourier transform by dividing the signal obtained over the entire time into specific time segments to incorporate time information. In this study, STFT calculations were performed through the 'signal' class in the TensorFlow library in Python (*TensorFlow. 2024.TensorFlow v2.13.0, 2023*). During this transformation, frame length is configured as 256, frame step as 160, and fft length as 384. Figure 1 displays the text associated with one example from each of the two datasets used in the study, along with the audio signal waveform and the spectrogram obtained through STFT.

### Language model

Supporting ASR systems, particularly for agglutinative languages like Turkish, with a language model is crucial for improving model performance. The deep learning approaches to be developed in the study were seq2seq models predicting character sequences from audio files. In this context, incorrect character predictions during the transcription of an audio file can result in the generation of incorrect words. Employing a language model offers the potential to normalize inaccurately produced words and generate meaningful results. The language model used in this study was developed by using a customized version of the Zemberek library, specifically designed for the Python language and tailored for Turkish (*Zemberek-NLP, 2023*). The Zemberek library, designed for Turkish natural language processing, encompasses operations such as Turkish morphology analysis, tokenization, and normalization. The primary reasons for using the Zemberek library in this study are its ease of use, open-source accessibility, and high success rate (*Kalender & Korkmaz, 2017*).

### Hyper-parameter optimization

In machine learning, hyperparameters are values that cannot be learned during training but play a pivotal role in controlling the training process. Consequently, they significantly influence the performance of machine learning models. Given that deep learning models encompass more hyperparameters compared to classical machine learning methods, the precise selection of hyperparameter values becomes paramount. Traditional hyper-parameter optimization methods like grid search are time-consuming and limited in their search space, making them ineffective for optimizing hyper-parameters in deep learning approaches. The hyper-parameters of the deep learning methods in this study were determined through Bayesian optimization, which demonstrated superiority over traditional methods (*Jones, 2001*; *Wu et al., 2019*; *Görmez & Aydin, 2023*). This method can quickly find the optimum hyper-parameter values by searching a wider space. The Bayesian optimization approach in this study was implemented using the 'skopt' library in Python (*Keras, 2023*). In this library, Gaussian process regression was implemented in gp_minimize function, in which acq_func was set to ''EI'', n_cals was set to 250 and all the other parameters were assigned to their default values.

### Proposed deep learning models

In this study, four models were created by using CNNs, LSTMs, Transformers, and GRU layers. In the first one of these models, the CNN layers connected immediately after the input layer were followed by the GRU layers in series. The number of CNN and GRU layers in this model showed variety and was optimized during the optimization phase. Following these layers, the model was completed with a fully connected and an output layer. The second model was identical to the first model, except that LSTM layers were used instead of GRU layers. In the third model, custom transformer modules were connected to the input layer, and the model was completed with a fully connected layer and an output layer. In the last model, akin to the first two models, CNN layers were connected sequentially to the input layer, followed by the addition of GRU, LSTM, and Transformer layers in parallel. Subsequently, the parallel connected layers were concatenated, and the model was finalized with a fully connected layer and an output layer.

The first model is named DeepTurkish_GRU, the second is DeepTurkish_LSTM, the third is DeepTurkish_Transformer, and the fourth is DeepTurkish_Hybrid. Figure 2 illustrates the architecture of the four models proposed in the study.

The GRU, LSTM, and Transformer layers in the models depicted in Fig. 2 are identical to each other. The number of layers and their hyper-parameters in each model are optimized individually. In each of the CNN modules in these models, batch normalization and ReLU layers are connected to two-dimensional convolution layers with kernel dimensions of $11 \times 41$. GRU and LSTM modules were developed as bidirectional with ReLU activation functions and supported by the Dropout layer. The transformer module comprises three layers: embedding, encoder, and decoder. The embedding layer consists of three 1-D convolution layers with sequential kernel sizes of 15, 11, and 9. In the Encoder and Decoder layers, there are MultiHeadAttention, Dense, LayerNormalization, and Dropout layers, respectively. While developing the transformer module, these layers were combined similarly to the architectural structure of the model proposed by *Dong, Xu & Xu (2018)*. In the classification layer of the model, there are 30 neurons (corresponding to the number of characters in the Turkish alphabet and the space character), and a fully connected layer with a softmax activation function is used. In our models, we chose the Adam optimizer, and the loss function was set as CTC, which has shown greater success than other loss functions in ASR.

## Turkish automatic speech recognition system

In this study, an end-to-end Turkish automatic speech recognition system is proposed, which includes feature extraction using spectrograms, character sequence prediction using deep learning, and text normalization using a language model. The flow diagram of the proposed model is shown in Fig. 3.

In the initial stage of the system depicted in Fig. 3, spectrograms of the audio data were obtained through STFT, as described in the feature extraction section. The spectrograms obtained at this stage served as inputs for the proposed deep learning models, which predicted the character sequences present in the audio data. A high WER in the prediction scores obtained from deep learning models is a common issue, particularly for agglutinative languages like Turkish (*Arslan & Barışcı, 2020*). Although CER is lower than WER, significant improvements in WER can be achieved by using a good language model. Hence, in the final stage of the proposed system, the language model developed by using Zemberek was incorporated, facilitating sentence normalization to produce audio text data.

## RESULTS

### Dataset preparation

In the initial stage of the study's experimental section, the datasets were partitioned to create training, testing, and validation datasets. The TSC dataset has already been divided into training, testing, and validation sets (*Mussakhojayeva et al., 2023*). In this context, there are 179,258, 3,484, and 3,428 samples in the training, testing, and validation datasets, respectively. The METU-1.0 dataset is provided as a single entity without such distinctions

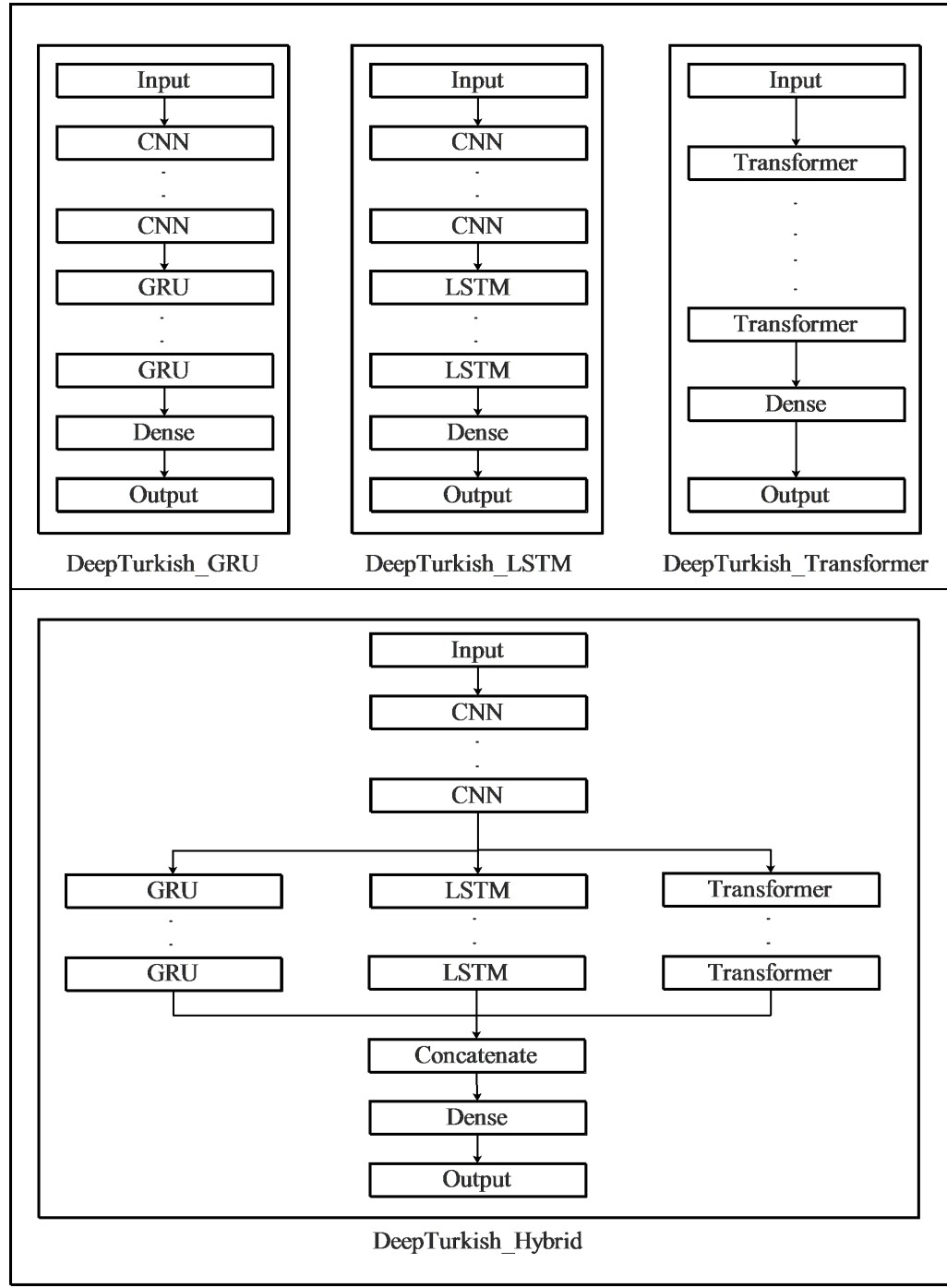

**Figure 2** Architecture of proposed deep learning models to use in Turkish automatic speech recognition system.

(*Salor et al., 2002*; *Salor et al., 2007*). For this reason, 20% of the METU-1.0 dataset was randomly selected to create the test dataset, and an additional 10% was chosen for the validation dataset, with the training dataset comprising the remaining samples. At the end

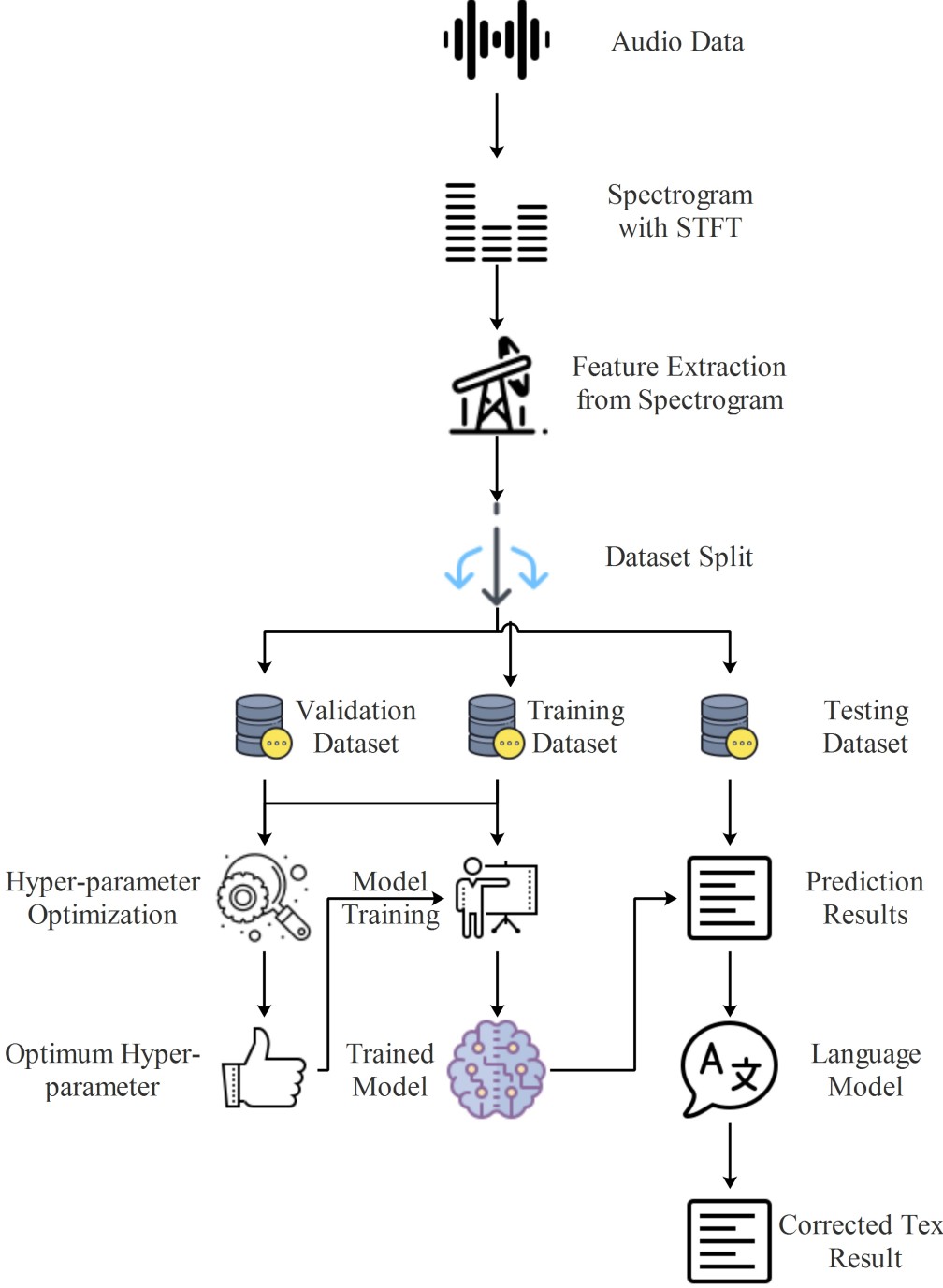

**Figure 3** **Flow diagram of proposed Turkish automatic speech recognition system.**

of this stage, the METU-1.0 dataset had 3,338 samples in the training dataset, 954 samples in the testing dataset, and 477 samples in the validation dataset. The training dataset was used to train the model, and the test dataset was used to evaluate the performance of the trained models. The validation dataset served two purposes: as test data during hyper-parameter

optimization and as validation data for callback functions during model training. After the optimization phase, the validation and training datasets were combined to train the master model.

The primary objective of the models developed within the study's scope is to extract character sequences from the input audio files. Turkish language comprises lowercase and uppercase versions of 29 distinct letters. To minimize class diversity, all characters were transformed to lowercase during the preprocessing phase. Furthermore, the texts from the audio files underwent cleaning and elimination of punctuation marks irrelevant to prediction within the proposed model. The non-Turkish characters within the text files were removed. Ultimately, the model was complemented by the Embed layer, aiding in the conversion of audio and text files into vectors.

## Experiments for hyper-parameter optimization

After the dataset preparation phase, the hyper-parameters of the four proposed deep learning models were individually optimized as described in the hyper-parameter optimization section. At this juncture, optimization was directed towards both layer-specific hyper-parameter values and artificial neural network-specific parameters, which play a pivotal role in the learning process. This encompassed the optimization of various aspects: the number of filters for CNN layers, the output space size for GRU and LSTM layers, the number and size of attention heads within the MultiHeadAttention layer of the Transformer module, the number of neurons in the fully connected layer, as well as the learning rate and number of epochs. Furthermore, the quantities of CNN, LSTM, GRU, and transformer layers were also optimized to ascertain the optimal depth of the model. Unlike grid search, the skopt library uses Gaussian processes to determine the optimal hyper-parameter values by considering the hyper-parameter's lowest value, highest value, and type. Table 1 represents the hyper-parameter name, type, search space and optimum value for each dataset of each model.

The values in Table 1 represent the best-performing hyper-parameter settings on the validation datasets for models trained on the training datasets. The best performance was determined by minimizing the average of WER and CER scores. Before calculating the WER and CER scores, the predicted character sequences were normalized through the language model to best represent the main model during the optimization phase.

When evaluating the results obtained, it is observed that deeper models generally demonstrate higher performance. Additionally, the number of units in the layers utilized tends to be closer to the maximum values within the hyper-parameter space. The variance in optimal values between the METU and TSC datasets is presumed to be correlated with the dataset sizes. With larger datasets, models tend to be deeper, and the number of units tends to be higher.

## Training and testing the deep learning models

The ultimate model was achieved through training it with the hyperparameter values determined during the optimization stage. In deep learning, model performance is directly related to the sample size, with larger samples resulting in improved performance. In

**Table 1  Hyper-parameter name, type, search space and optimum value for each dataset of each model.**

| Hyper-parameter name | Hyper-parameter type | Lowest value for search space | Highest value for search space | Model name | Dataset | Optimum value |
|---|---|---|---|---|---|---|
| Number of CNN module | Integer | 1 | 10 | DeepTurkish_GRU | METU-1.0 | 8 |
| | | | | | TSC | 8 |
| | | | | DeepTurkish_LSTM | METU-1.0 | 6 |
| | | | | | TSC | 7 |
| | | | | DeepTurkish_Hybrid | METU-1.0 | 1 |
| | | | | | TSC | 5 |
| Number of filters for CNN | Integer | 32 | 256 | DeepTurkish_GRU | METU-1.0 | 171 |
| | | | | | TSC | 181 |
| | | | | DeepTurkish_LSTM | METU-1.0 | 73 |
| | | | | | TSC | 197 |
| | | | | DeepTurkish_Transformer | METU-1.0 | 96 |
| | | | | | TSC | 168 |
| | | | | DeepTurkish_Hybrid | METU-1.0 | 154 |
| | | | | | TSC | 215 |
| Number of GRU module | Integer | 1 | 10 | DeepTurkish_GRU | METU-1.0 | 6 |
| | | | | | TSC | 9 |
| | | | | DeepTurkish_Hybrid | METU-1.0 | 6 |
| | | | | | TSC | 7 |
| Number of GRU units | Integer | 32 | 256 | DeepTurkish_GRU | METU-1.0 | 250 |
| | | | | | TSC | 190 |
| | | | | DeepTurkish_Hybrid | METU-1.0 | 164 |
| | | | | | TSC | 47 |
| Number of LSTM module | Integer | 1 | 10 | DeepTurkish_LSTM | METU-1.0 | 10 |
| | | | | | TSC | 1 |
| | | | | DeepTurkish_Hybrid | METU-1.0 | 7 |
| | | | | | TSC | 6 |
| Number of LSTM units | Integer | 32 | 256 | DeepTurkish_LSTM | METU-1.0 | 227 |
| | | | | | TSC | 64 |
| | | | | DeepTurkish_Hybrid | METU-1.0 | 159 |
| | | | | | TSC | 219 |
| Number of transformer module | Integer | 1 | 10 | DeepTurkish_Transformer | METU-1.0 | 1 |
| | | | | | TSC | 1 |
| | | | | DeepTurkish_Hybrid | METU-1.0 | 2 |
| | | | | | TSC | 5 |
| Number of attention heads. | Integer | 1 | 10 | DeepTurkish_Transformer | METU-1.0 | 9 |
| | | | | | TSC | 10 |
| | | | | DeepTurkish_Hybrid | METU-1.0 | 4 |
| | | | | | TSC | 8 |
| Size of each attention head for query and key | Integer | 32 | 256 | DeepTurkish_Transformer | METU-1.0 | 76 |
| | | | | | TSC | 181 |
| | | | | DeepTurkish_Hybrid | METU-1.0 | 139 |
| | | | | | TSC | 33 |

**Table 1** (*continued*)

| Hyper-parameter name | Hyper-parameter type | Lowest value for search space | Highest value for search space | Model name | Dataset | Optimum value |
|---|---|---|---|---|---|---|
| Number of neurons in Dense Layer | Integer | 32 | 256 | DeepTurkish_GRU | METU-1.0 | 233 |
| | | | | | TSC | 100 |
| | | | | DeepTurkish_LSTM | METU-1.0 | 157 |
| | | | | | TSC | 120 |
| | | | | DeepTurkish_Transformer | METU-1.0 | 127 |
| | | | | | TSC | 105 |
| | | | | DeepTurkish_Hybrid | METU-1.0 | 147 |
| | | | | | TSC | 243 |
| Initial Learning Rate | Real | $10^{-4}$ | $10^{-1}$ | DeepTurkish_GRU | METU-1.0 | 0.008493 |
| | | | | | TSC | 0.042409 |
| | | | | DeepTurkish_LSTM | METU-1.0 | 0.088681 |
| | | | | | TSC | 0.077663 |
| | | | | DeepTurkish_Transformer | METU-1.0 | 0.040433 |
| | | | | | TSC | 0.027824 |
| | | | | DeepTurkish_Hybrid | METU-1.0 | 0.002753 |
| | | | | | TSC | 0.060049 |
| Number of epoch | Integer | 100 | 1,500 | DeepTurkish_GRU | METU-1.0 | 607 |
| | | | | | TSC | 774 |
| | | | | DeepTurkish_LSTM | METU-1.0 | 1,062 |
| | | | | | TSC | 1,177 |
| | | | | DeepTurkish_Transformer | METU-1.0 | 1,169 |
| | | | | | TSC | 1,027 |
| | | | | DeepTurkish_Hybrid | METU-1.0 | 565 |
| | | | | | TSC | 958 |

this context, the validation and training datasets were concatenated to increase the sample size for training the final model. In addition to this, the aim was to enhance the model's performance by incorporating two callback functions, namely *lr_callback* and *early_stopping_callback*, into the model. When the loss value does not improve consecutively for two iterations during model training, the *lr_callback* reduces the learning rate by half. Furthermore, if there is no improvement for six consecutive iterations, the *early_stopping_callback* terminates the model training. The Keras library in Python was utilized to develop all the model and callback functions (*Keras, 2023a*). All model settings, except for hyper-parameters, were configured in accordance with the specifications outlined in the proposed deep learning models section, leaving the remaining parameters at their default values in the Keras library. Upon completing the model training, the trained model was used to predict the character sequences for each sample in the test dataset. For each model and dataset, two sets of WER and CER scores were computed: one based on the raw predictions and another after passing them through the language model. The WER and CER scores, calculated for each model and dataset, are presented in Table 2.

**Table 2  WER and CER scores of proposed models computed using METU-1.0 and TSC datasets for both models with language model and without language model.**

| Model name | Without language model | | | | With language model | | | |
|---|---|---|---|---|---|---|---|---|
| | METU-1.0 | | TSC | | METU-1.0 | | TSC | |
| | WER | CER | WER | CER | WER | CER | WER | CER |
| DeepTurkish_GRU | 27.3 | 15.80 | 13.1 | 6.05 | 10.50 | 6.85 | 9.2 | 3.30 |
| DeepTurkish_LSTM | 26.1 | 15.65 | 12.5 | 5.85 | 10.12 | 6.30 | 9.0 | 3.05 |
| DeepTurkish_Transformer | 24.3 | 15.45 | 12.6 | 5.95 | 9.92 | 5.45 | 8.6 | 2.80 |
| DeepTurkish_Hybrid | 22.2 | 14.95 | 11.5 | 4.15 | 9.85 | 5.35 | 8.4 | 2.70 |

**Table 3  Sentence predictions by model DeepTurkish_Hybrid and actual sentences for randomly selected samples.**

| Actual sentence | English version of actual sentence | Predicted sentence without language model | CER/WER | Predicted sentence with language model | CER/WER |
|---|---|---|---|---|---|
| Ona bir patlattıve karanlığın içine düştü | He blasted her and she fell into the darkness | Ona bir patkatıve kaaanlığın içine düştü | 0.071/0.285 | Ona bir patlattıve karanlığın içine düştü | 0.0/0.0 |
| Genellikle kırıntılarıdenize atarlardı | They usually threw the crumbs into the sea | Gene kimle kırıntılarıdeniz atarlar | 0.153/1.0 | Gene kimle kırıntılarıdeniz atarlar | 0.153/1.0 |
| Deniz niye öbürlerinin gitmesine izin versin ki | Why would Deniz let the others go | Deniz niye öbürlerinin gitmesine izin versin ki | 0.0/0.0 | Deniz niye öbürlerinin gitmesine izin versin ki | 0.0/0.0 |
| Ama sadece bu bölümde dinleyicileri aldık | But we only got listeners in this episode | Ama sadece bu bölümde dinde içleri aldık | 0.097/0.333 | Ama sadece bu bölümde dinde içleri aldık | 0.097/0.333 |
| Bu yeni yöntemleri günlük hayatta kullanmak son basamak | Using these new methods in daily life is the last step. | Bu yeni yöntemleri günülük ayakta kullanmak son basamak | 0.054/0.250 | Bu yeni yöntemleri günlük ayakta kullanmak son basamak | 0.036/0.125 |

When examining the results in Table 2, it becomes evident that the model utilizing a hybrid of CNN, LSTM, GRU, and Transformer layers achieves the best performance. As expected, the WER score is higher than the CER score for all models. Another inference drawn from Table 2 is the superior performance of approaches integrating language models compared to those that do not. Establishing the statistical significance of this enhancement is crucial for validating the efficacy of the language model. Consequently, employing the outcomes derived from the best-performing hybrid model, a two-tailed $Z$-test was conducted between approaches utilizing a language model and those that do not. The resulting two-tailed $Z$-test outcomes indicate statistically significant improvements for both CER and WER scores in both the METU-1.0 and TSC datasets, meeting at $p < 0.01$ threshold. Hence, it is deduced that the incorporation of a language model is imperative for the design of an effective ASR system. A character mistake can significantly alter the meaning of words that share a similar character sequence. Hence, analyzing predicted sentences alongside actual sentences provides valuable insights into model performance. Table 3 displays predicted sentences by the DeepTurkish_Hybrid model alongside the actual sentences for randomly selected samples.

The results in Table 3 shows that the model achieved perfect accuracy in predicting some sentences while making minor errors in others. It is evident that a sentence can be easily corrected by the language model if only a few characters within a word are mistaken.

However, the results indicate that there are instances where the language model cannot correct certain errors. The primary reason for this phenomenon is that the error in the word predicted by deep learning is not limited to a character mistake alone. The incorrectly predicted word is a valid word in the Turkish language, so the proposed language model does not make any corrections. While such minor errors were not common, they hold significant importance because they can alter the meaning of words. Therefore, correcting these errors is essential for system performance. The examination of the WER and CER scores calculated for each example proves that it is beneficial in discerning the impact of the language model's enhancement. Upon review, it becomes evident that the WER and CER scores attained by approaches employing a language model are equal to or superior to those achieved by approaches lacking a language model. Particularly in certain instances, the utilization of a language model has the potential to reduce WER and CER scores to 0. In several cases, substantial improvements have been observed, even if the scores are not reduced to zero.

## DISCUSSION

Through the literature research conducted in this study, it has been found that deep learning methods are commonly employed in recent studies on ASR systems (*Abdel-Hamid et al., 2014*; *Wang, Wang & Lv, 2019*; *Oruh, Viriri & Adegun, 2022*). It has been observed that these studies commonly incorporate layers such as GRU, LSTM, and Transformer, resulting in the most successful outcomes among the methods utilized (*Chiu et al., 2018*; *Liu et al., 2018*; *Hsu et al., 2021*; *Yu et al., 2022*). It has been noted that a majority of ASR studies in existing literature primarily concentrate on languages such as English, Chinese, Spanish, German, and French, whereas there is a scarcity of research focusing on languages within the agglutinative language group, such as Turkish (*Arslan & Barışcı, 2020*).

In this study, we developed a seq2seq deep learning method for Turkish ASR systems, which was further enhanced by incorporating a language model. Upon reviewing the obtained results, it was evident that the best-performing model was DeepTurkish_Hybrid. This model utilized a hybrid approach by combining CNN, LSTM, GRU, and Transformer layers. When the DeepTurkish_Hybrid method was complemented with our proposed language model, we achieved WER scores of 9.85 and 8.4, as well as CER scores of 5.35 and 2.7 for the TSC and METU-1.0 datasets, respectively.

Based on the obtained results, it has been observed that the model has demonstrated excellent performance in Turkish automatic speech recognition. Upon reviewing the actual sentences from the voice data within the datasets and comparing them to the sentences predicted by the model, it becomes apparent that the system, when operating without a language model, occasionally makes minor character errors. These errors are largely rectified by the language model; however, in some instances, character errors have led to the generation of words that already exist in the Turkish language. In these rare cases, the language model was unable to correct the word. It was noted that the performance disparity between approaches employing and not employing a language model, concerning both CER and WER scores, exhibited significance at the threshold of $p < 0.01$. This observation

**Table 4  State-of-the-art comparison of DeepTurkish_Hybrid with respect to the WER score.**

| Model | Dataset | WER score |
|---|---|---|
| Mussakhojayeva et al. | TSC | 9.6 |
| Oyucu and Polat | METU-1.0 | 61.9 |
| Tombaloğlu and Erdem | METU-1.0 | 10.65 |
| Ciloglu et al. | METU-1.0 | 35.91 |
| DeepTurkish_Hybrid | TSC | 8.40 |
| | METU-1.0 | 9.85 |

was confirmed by the two-tailed $Z$ test conducted on both datasets. Considering all these findings, it is anticipated that the proposed model can be seamlessly integrated into various ASR systems.

While rare, there are cases where the deep learning method predicts an incorrect word, and the language model fails to correct it, contributing to higher WER and CER scores for the model. Therefore, minimizing such instances is crucial for enhancing model performance. It is anticipated that these incorrectly predicted sentences by the proposed system can be effectively corrected with adjustments to the language model. For example, situations like the sentence actually being '*Ama sadece bu bölümde dinleyicileri aldık*' being predicted as '*Ama sadece bu bölümde dinde içleri aldık*' could potentially be addressed with a language model capable of sentence analysis. This is important as,- , the sentence lacks semantic coherence despite containing words entirely in Turkish. Consequently, incorporating a robust language model capable of sentence analysis into our model design is expected to significantly improve its success rate. To achieve this, we aim to develop a powerful language model using data collected in future studies.

To assess the validity of the proposed models more effectively, it is important to compare them with other studies in the literature based on their performance. The studies in the literature commonly use the WER score as the primary criterion for evaluating ASR systems. Hence, we compared DeepTurkish_Hybrid, which achieved the best result among our proposed models, with existing studies in terms of the WER score. In this context, the model trained on the TSC dataset was compared with the study by *Mussakhojayeva et al. (2023)*, who prepared the dataset. The model trained using the METU-1.0 dataset was compared with studies of *Oyucu & Polat (2023)* (*Tombaloğlu & Erdem, 2021*; *Ciloglu, Comez & Sahin, 2004*; *Tombaloğlu & Erdem, 2021*). The comparison results are presented in Table 4, and based on these results, our model outperformed all of the mentioned studies.

*Oyucu & Polat (2023)* utilized a 3-layer LSTM and a skip-gram based language model using Kaldi technology. *Tombaloğlu & Erdem (2021)* leveraged Kaldi technology to support a deep learning model based on LSTM and GRU, complemented by a sub-word based language model. *Ciloglu, Comez & Sahin (2004)* employed their proposed language model in their research. *Mussakhojayeva et al. (2023)* trained a Transformer based deep model for multilingual purposes in their study. In our work, a novel deep learning model that simultaneously incorporates LSTM, GRU, and Transformer layers was proposed.

The depths and hyper-parameters of these models have been optimized. Our model is augmented with a Zemberek-based language model. These features distinguish our proposed model from the comparison models which were shown in Table 4.

According to the results in Table 4, it becomes evident that the proposed model outperforms those documented in the literature. The model closest in performance to the proposed model for the METU-1.0 dataset was the one presented by *Tombaloğlu & Erdem (2021)*. Nevertheless, the model introduced in this study achieved a WER score 0.8 points lower than this aforementioned model. Similarly, concerning the TSC dataset, the proposed model attained a WER score 1.2 points lower than the model proposed by *Mussakhojayeva et al. (2023)*. Based on these findings, it is anticipated that the proposed model would be a more favorable choice than existing models in the literature for the development of a Turkish ASR system. It is believed that the discrepancies within the proposed model account for its superior performance compared to other models documented in the literature. Specifically, conducting a systematic hyper-parameter optimization, devising a novel deep learning model from scratch, and incorporating a language model are considered to positively impact the model's performance, thus enabling it to outperform existing models in the literature.

## CONCLUSIONS

ASR problems are recognized to present unique challenges compared to other problems such as classification, which often results in more complex deep learning models tailored for ASR. Notably, ASR systems designed for the agglutinative language group, such as Turkish, face additional complexities compared to the models in other languages. Thus, we developed and applied various deep learning models with different layers to Turkish datasets in this context. Hyper-parameter optimization is an important process for deep learning models, so the hyper-parameters of the deep learning models proposed in our study were optimized through the Bayesian optimization method. The reason of using Bayesian optimization is that it is faster than other methods for deep learning models and can search in a larger space. Based on the experimental results, it became evident that among the deep learning models incorporating CNN, LSTM, GRU, and Transformer layers, the hybrid model utilizing all layers exhibited the highest performance. Furthermore, substantial improvements in success rates were observed when the proposed model was augmented with the language model. Upon comparing the achieved results with existing literature, it was apparent that our proposed model outperformed previous studies. However, it was also noted that there were some shortcomings on the language model dimension of the research. As a result, it is recommended that future studies prioritize the design and enhancement of the language model.

The results highlight numerous strengths of the study; however, there exist also certain limitations that should be acknowledged. One primary limitation is the scarcity of datasets available in the literature for the Turkish language. Deep learning methods typically exhibit a direct correlation between dataset size and model performance. As illustrated by the improved performance achieved by using the TSC dataset, which contains a greater number

of samples compared to the METU-1.0 dataset, superior performance can be attained to larger datasets. Therefore, it is hypothesized that reiterating the models proposed in the study which uses higher-dimensional data could enhance the performance scores. Another limitation of the study pertains to the perceived inadequacy of the employed language model in certain scenarios. The obtained results demonstrate the enhancement in ASR performance facilitated by the language model. Consequently, it is anticipated that employing a more robust language model could potentially augment the performance of the proposed models.

## ACKNOWLEDGEMENTS

The experiments reported in this article were performed at Tubitak Ulakbim, High Performance and Grid Computing Center (TRUBA resources).

### Funding
The author received no funding for this work.

### Competing Interests
The author declares there are no competing interests.

### Author Contributions
- Yasin Görmez conceived and designed the experiments, performed the experiments, analyzed the data, performed the computation work, prepared figures and/or tables, authored or reviewed drafts of the article, and approved the final draft.

### Data Availability
The METU-1.0 dataset is available at the Linguistic Data Consortium: Salor, Ozgul, et al. Middle East Technical University Turkish Microphone Speech v 1.0 LDC2006S33. Web Download. Philadelphia: Linguistic Data Consortium, 2006. https://doi.org/10.35111/sk8b-ss58.

The Turkish Speech Corpus is available at GitHub: https://github.com/IS2AI/TurkicASR/tree/main?tab=readme-ov-file.

The codes are available in the Supplementary Files.

### Supplemental Information
Supplemental information for this article can be found online at http://dx.doi.org/10.7717/peerj-cs.1981#supplemental-information.

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
