# Peer review of "Customized deep learning based Turkish automatic speech recognition system supported by language model"

_PeerJ Computer Science, doi:10.7717/peerj-cs.1981_

## Round 0.1 · original submission · Major Revisions

Please consider the comments of both reviewers.

Explain what makes Turkish language different/difficult than other languages?

What are the studies on similar languages?

Fix the citation issues (Salor et al., 2002, 2007)

Add more justification of the hyper-parameter

Explain the results in Tables 1& 2 well. Highlight/bold the best outcomes.

Figure 2 seems to be two figures? (model1-3: one figure & model 4: second figure)?

Revise the size of Figure 3, too big.

**Language Note:** The review process has identified that the English language must be improved. PeerJ can provide language editing services - please contact us at copyediting@peerj.com for pricing (be sure to provide your manuscript number and title). Alternatively, you should make your own arrangements to improve the language quality and provide details in your response letter. – PeerJ Staff

Reviewer 1 ·

Basic reporting

The results are very promising, and the application problem is important in big data problem-solving.
There are some major points which need to be improved.
1. The information given in Table 1 should be detailed.
2. The limitations of this study should be given.
3. The introduction should be divided into two and a literature review title.
4. The contributions of the work given in the introduction to this paper can be improved.
5. Please give some examples of the proposed method in the paper.
6. The novelty of the proposed method should be highlighted. The authors should clarify the paper's contributions in the introduction section.
7. In the introduction, the paper's motivation must be articulated far more clearly.
8. There are some spelling mistakes in the article. The article should be read from beginning to end.

Experimental design

.

Validity of the findings

.

Additional comments

no comment.

·

Basic reporting

The paper is scientifically sound and well written, with an exhaustive set of experiments on a suite of different scenarios. I see merit in this research paper.

In this paper, model of deep learning supported ASR system with language model. Experiments on specific four models and 2 datasets are conducted to evaluate and compare the method.

This paper is well organised. Ablation study is provided. The hyperparameter is provided, which is appreciated.


Questions:
1. at Line 112.

Based on the literature, it is evident that the existing ASR has some challenges. Given that these 113 challenges vary from one language to another, designing an ASR system, especially for 114 agglutinative languages like Turkish, is notably more complex compared to other languages. The 115 structure of the Turkish language allows for the generation of billions of different morphological 116 forms (Palaz et al., 2005).

This claim in introduction section is too light that should be supported the latest evidence of complexities in Turkish compared to other language.


2. The result in Table 2 and 3 need detail support for the validity of the result.

Experimental design

There are rigorous investigation performed based on technical standard.
However,
1. I could not find the justification of the hyper-parameters, the one that appeared was not that clear.
2. The preparation to the pre processing before hand is arguably being briefly explained.

Validity of the findings

3. For Table 2. this author should add a validity analysis to the possibility of the changes from no language model vs language model exist.
4. Table 3 result of prediction is seem unreadable for me, and need some explanation in English and the detail analysis.
5. The reference to the result should also be incorporated in discussion section.

Additional comments

no comment.

---

## Round 0.2 · Minor Revisions

Dear author,

The original Academic Editor is unavailable so I have taken over handling your submission.

The reviews for your manuscript have been received. Your paper needs some minor revisions. In particular, you are expected to revise the paper according to the basic reporting and validity of the findings of reviewer 2. In addition, the originality or novelty of the study should be mentioned. The main contributions should be listed in the Introduction section. The organization of the paper should be presented at the end of the Introduction section. Many sections are too long to read. They should be divided into two or more paragraphs to improve readability and comprehension.

Best wishes,

Reviewer 1 ·

Basic reporting

All the revisions are done. It looks great.

Experimental design

All the revisions are done. It looks great.

Validity of the findings

All the revisions are done. It looks great.

Additional comments

All the revisions are done. It looks great.

·

Basic reporting

The abstract and the introduction has enormously improved.

However, in other section there will be some query and improvement needed:
1) Give sufficient related work about Zemberek Model.
2) What is the summary for the related works the author has studied.
3) At line 160 and 169 - citation is not completed

How the method of feature extraction and Hyper-Parameter Optimization work in the phase of the project? how the methods works in terms of flow chart or mathematical formulation with explanation.

Experimental design

no comment.

Validity of the findings

In order to support your validity if finding, at line 429-436, show the similarity and the different among the related researcher and discuss it in summary.

In order to support the result in Table 1, show the meaning of the result such as discussion and interpretation the results/finding that reflect to the article's aim.

The discussion section described by recapping the material and method and slightly brief touch the finding based material and method used in the study.

Additional comments

The article can be accepted with minor revisions.

---

## Round 0.3 · Minor Revisions

Dear author,

The reviews for your revised manuscript have been received. Your paper still needs minor revision. You will be expected to revise the paper according to the experimental design comments provided by reviewer 2.

Best wishes,

**Language Note:** The review process has identified that the English language must be improved. PeerJ can provide language editing services - please contact us at copyediting@peerj.com for pricing (be sure to provide your manuscript number and title). Alternatively, you should make your own arrangements to improve the language quality and provide details in your response letter. – PeerJ Staff

·

Basic reporting

The authors present models on Deep Learning for automatic speech recognition systems for Turkish language model by integrating it with other language model. It is evaluated by the performance analysis based on word error rate.

Experimental design

The paper is scientifically sound and well written, with a set of experiments. I see merit in this research paper and I have only some points to be clarified:
i. The statement from line 56- 69 needed citation(s).
ii. I see that, line 229-230 also needed citation(s).
iii. The result in table 4 is not well explained and analysed.
iv. Show that line 459-468 are supporting table 4 or any other context they meant for..

Validity of the findings

Have been improved.

Additional comments

The authors need to give support to their statements especially in the mentioned statements.
The article also need proofreading.

---

## Round 0.4 · accepted · Accept

Dear authors,

Thank you for the revision and for addressing all the reviewers' comments. I confirm that the paper is improved and clearly addresses the concerns of the reviewers. Your paper is now acceptable for publication in light of this last revision.

Best wishes,

·

Basic reporting

The authors present models on Deep Learning for automatic speech recognition systems for Turkish language model by integrating it with other language model. It is evaluated by the performance analysis based on word error rate.

Experimental design

no comment.

Validity of the findings

The authors manages to validate their findings based on previous query/comments and rigorous outlined their finding and analysis.

Additional comments

The authors managed to prove that their proposed algorithm achieved better results than the reviewed related works.



The paper is scientifically sound and well written, with a set of experiments. I see merit in this research paper.